# Effect of Cephalosporin Antibiotics on the Activity of Yoghurt Cultures

**DOI:** 10.3390/foods11182751

**Published:** 2022-09-07

**Authors:** Pavlina Navrátilova, Ivana Borkovcova, Zora Stastkova, Ivana Bednarova, Lenka Vorlova

**Affiliations:** Faculty of Veterinary Hygiene and Ecology, University of Veterinary Sciences Brno, Palackeho tr. 1946/1, CZ-612 42 Brno, Czech Republic

**Keywords:** dairy cultures, fermentation process, antibiotics, cephalosporin’s, maximum residue limit

## Abstract

The presence of antibiotics in milk is a significant problem affecting the technological safety of dairy products. The aim of the study was to determine the sensitivity of yoghurt cultures to residual levels of selected cephalosporin antibiotics (cephalexin, cefoperazone, cefquinome, cefazolin, and ceftiofur). Five yoghurt cultures were selected containing strains of *Lactobacillus delbrueckii* subsp. *bulgaricus* and *Streptococcus thermophilus*. Artificially fortified milk samples (whole pasteurized milk; 85 °C; 3–5 s) with cephalosporins at a concentration of the maximum residue limit were used to evaluate the sensitivity of the yoghurt cultures by monitoring the pH, titratable acidity, and the concentration of selected organic acids (lactic, pyruvic, citric, acetic, orotic, oxalic, formic, uric, and succinic acids) at the end of fermentation (43 °C; 4–5.5 h; pH ≤ 4.6). The titratable acidity was determined by the Soxhlet–Henkel method and the organic acid concentration was monitored by reversed-phase HPLC. Ceftiofur had the greatest effect on the yoghurt culture activity, with a statistically highly significant effect (*p* < 0.05) on the pH, titratable acidity, and the content of lactic, pyruvic, and acetic acids in all cultures. Other cephalosporins also showed an inhibitory effect on yoghurt metabolism as seen by the evaluation of the lactic and pyruvic acid concentrations.

## 1. Introduction

The presence of veterinary pharmaceutical residues in milk is a significant problem for ensuring the safety of raw materials for the manufacture of dairy products [1]. The drugs (antibiotics, chemotherapeutics, etc.) used to treat lactating dairy cows can pass into milk. The most important source of antibiotic residues in milk is their use in mastitis therapy and in the dry period. Residues of veterinary drugs in milk, especially when present at concentrations exceeding permitted limits (maximum residue limit—MRL), may pose a health risk to the consumer [2].

The main task for food processors is to ensure the safety of the final products. This cannot be achieved without ensuring the technological safety of the raw materials. The presence of antibiotics in milk is a significant problem affecting the technological safety of dairy products. It has been known for a long time that antimicrobial residues in milk can significantly affect production when using dairy cultures. The problem of antibiotics in milk was pointed out as early as 1948 by Kastli, who reported that milk from udders treated with penicillin could interfere with the manufacture of dairy products. He showed that penicillin residues at concentrations of 0.1–1 IU (international units) inhibited the growth of starter cultures [3]. This publication was the impetus for a series of studies aimed at determining the effect of antibiotics on the activity of dairy cultures and on the quality of dairy products. This subsequently started a broad discussion in the scientific and professional literature, which continuesto this day as shown by a number of scientific works [4,5,6,7,8,9,10,11]. These studies have shown that the presence of antibiotic residues in raw milk causes problems during milk processing. Milk’s contamination with antibiotic residues can lead to a decrease in acidityand to the occurrence of undesirable sensory changes in fermented dairy products, reduce curdling, negatively impact cheese ripening, and affect the growth and metabolism of cultures used in the dairy industry [6,12]. Data on the sensitivity of dairy cultures to antimicrobial agents often vary in the literature, because even individual strains of the same species of lactic acid bacteria can show different sensitivities. The factors affecting the sensitivity of pure lactic acid cultures include the type of culture, the composition of the culture (monoculture x mixed culture), and the type of antimicrobial (the mechanism of action of the antibiotic on the microbial cell) [7,13]. Apart from changes in active and titratable acidity, the determination of organic acid concentrations can be used to monitor growth and the metabolic activity of starter cultures [10,14,15]. Organic acids have a diverse origin and are mainly formed when lactic acid bacteria metabolize carbohydrates (lactic, acetic, pyruvic, propionic, and formic acids), or are products of milk fat hydrolysis (butyric acid). Some occur naturally as products of physiological metabolism (citric, orotic, and uric acids) in milk or products of bacterial metabolism during fermentation [16].

Although cephalosporins are β-lactam antibiotics and have been regularly used to treat mastitis, studies dealing with the effect of cephalosporins on the growth and metabolism of dairy cultures are rather scarce [9,10]. Cephalosporins are antibacterial drugs with a broad spectrum of action [8], and the use of 3rd and 4th generation cephalosporins has been associated with a risk of the selection of bacteria carrying *ESBL* (extended-spectrum *β*-lactamase) and *AmpC* (cephalosporinase) genes in treated animals. Hence, the 3rd and 4th generation cephalosporins are classified as critically significant antimicrobials and indicatory limits have been imposed [17]. Their consumption trends are being monitored with increasing intensity (e.g., through evaluations and consumption comparisons between EU member states). Pokludova et al. [18] state that the administration of this group of substances to dairy cattle should be carefully managed, given that veterinary pharmaceuticals, including 3rd and 4th generation cephalosporins, are being overused, especially with regard to certain indications. Veterinary pharmaceuticals registered in the Czech Republic for bovines include 1st, 3rd, and 4th generation cephalosporins (cephalexin, cefazolin, cefquinome, ceftiofur, cefoperazone, cefapirin, cefalonium, and cefacetril), most of which are intended for intramammary administration during lactation and dry periods. Analyses of the consumption of veterinary medicines in the Czech Republic from 2010–2017 show no reduction in the consumption of these substances, which also have a highly significant impact on human medicine.

According to a survey by the Federation of Veterinarians of Europe (FVE), some Generation III and IV cephalosporins have a zero-withdrawal period for milk, and are overused in lactating cattle, often without a clear diagnosis [19]. The question is whether the MRLs set for the cephalosporins in milk represent levels that do not have inhibitory effects on dairy cultures. According to the EU legislation (Commission Regulation (EU) No 37/2010) [20], the MRLs for cephalosporins in milk are as follows: cephalonium 20 μg kg^−1^, cefapirin 60 μg kg^−1^, cephalexin 100 μg kg^−1^, ceftiofur 100 μg kg^−1^, cefoperazone 50 μg kg^−1^, cefquinome 20 μg kg^−1^, and cefazolin 50 μg kg^−1^. The application for marketing authorization of a veterinary medicinal product (VMP) must contain data and documents required by EU legislation [21]. The dossier must contain data relating to the safety of the veterinary medicinal product and its residues. Among other effects, the safety tests focus on the possible harmful effects of VMP residues in food for human consumption arising from treated animals, as well as on the problems that these residues may cause in industrial food processing [21].

The aim of our study was to determine the effect on the activity of yoghurt cultures of the residues of selected cephalosporin antibiotics (cephalexin, cefoperazone, cefquinome, cefazolin, and ceftiofur) at a concentration of 1 MRL on milk. Five thermophilic yoghurt cultures were selected containing *Lactobacillus delbrueckii* subsp. *bulgaricus* and *Streptococcus thermophilus*, strains typical for yoghurt culture. The activity of the cultures was evaluated by monitoring active and titratable acidity and by measuring the concentration of organic acids (lactic, pyruvic, citric, acetic, orotic, oxalic, formic, and uric and succinic) at the end of fermentation.

## 2. Materials and Methods

### 2.1. Yoghurt Cultures

Five thermophilic yoghurt cultures containing strains typical of the yoghurt culture *Lactobacillus delbrueckii* subsp. *bulgaricus* and *Streptococcus thermophilus* were employed. Four yoghurt cultures—FD-DVS YC-X11 (designated as culture Z), FD-DVS YF-L903 (culture Y), FD-DVS YF-L812 Yo Flex (culture X), and FD-DVS YC-381 YoFlex (culture W)—were obtained from Christian Hansen (Chr. Hansen Holding A/S, Hoersholm, Denmark) and DELVO YOG CY-340-DSL (culture V) was obtained from DSM Food Specialties (Koninklijke DSM N.V., Heerlen, The Netherlands).

### 2.2. Antibiotics and Chemicals

The following analytical standards of cephalosporin antibiotics were used: Cefoperazone sodium (CMS8039, HiMedia Laboratories, Pvt. Ltd., Mumbai, India), Cephalexin (CMS647, HiMedia), Ceftiofur sodium (PHR1521, Sigma Aldrich Inc., St. Louis, MA, USA), Cephazolin (CMS 650 HiMedia), and Cefquinome sulphate (32472 Sigma Aldrich).

Analytical standards of organic acids were purchased from Sigma Aldrich: formic acid ACS reagent ≥96% (695076), pyruvic acid (107360), orotic acid monohydrate 97% (08402), lactic acid 85% FCC (W 261106), uric acid ≥99% (U2625), and succinic acid p.a. ACS reagent ≥99.5% (14079). Citric acid monohydrate p.a. (CAS 5949-29-1), oxalic acid dihydrate p.a. (CAS 6153-56-6), potassium hexacyanoferrate (II) trihydrate (CAS 14459-95-1), zinc sulfate heptahydrate (CAS 7446-20-0), phenolphthalein indicator (CAS 77-09-8), and sodium hydroxide (CAS 1310-73-2) were purchased from Penta s.r.o. (Prague, Czech Republic).

### 2.3. Determination of Active and Titratable Acidity

The pH (active acidity) values were determined with a microprocessor pH-Meter (Model Hanna pH 210) fitted with a probe (Model THETA HC 144). Prior to the measurements, the apparatus was calibrated at pH 4 and pH 7.

The titratable acidity of milk and yoghurt determined by the Soxhlet–Henkel method is given by the number of milliliters of 0.25 mol L^−1^ NaOH solution consumed in titrating 100 mL of sample with phenolphthalein (c = 2 *w*/*w*) as indicator. The results are expressed as Soxhlet–Henkel degrees (°SH).

### 2.4. Preparation of Yoghurt Culture

Cultures were stored at −18 °C. The contents of the package were weighed and divided into 5 equal parts. A weighed amount of culture was dissolved under sterile conditions in a small amount of pasteurized milk in a 500 mL volumetric flask, and the volume was made up to the mark. Depending on the recommended inoculation ratio, the inoculation dose (in mL) was determined. The inoculation ratio was 250 L/50 U (units) for cultures Z, Y, X, and W, and 500 L/2 U for culture V.

### 2.5. Preparation of Milk Samples Containing Cephalosporins

Working solutions of antibiotics at a concentration of c = 0.01 mg mL^−1^ were prepared from analytical standards of antibiotics by dilution with distilled water. Raw whole cow’s milk was high-temperature pasteurized (85 °C, 3–5 s), cooled to 20–25 °C, and a calculated amount of cephalosporin antibiotic working solution was added to reach a concentration of 1 MRL. This medium was then inoculated with yoghurt culture, mixed thoroughly, and the milk samples were divided into sterile beakers containing 50 mL each. The beakers were sealed with aluminum foil and placed in a water bath at 43 °C. A blank sample without the addition of antibiotic was prepared in parallel. A total of 8 blanks and 8 samples were prepared each time for each cephalosporin and each culture.

### 2.6. Determination of Yoghurt Culture Activity

Fermentation was terminated when pH ≤ 4.6 in the blank. The samples were cooled to laboratory temperature and the activity of the yoghurt cultures was monitored by measuring the acidity (pH and titratable acidity) of the blank and the samples.

### 2.7. Analysis of Organic Acids by HPLC

The activity of the cultures was also assessed by determining the concentration of selected organic acids. A total of 10 g of milk or 5 g of yoghurt was weighed into a 50 mL volumetric flask. A total of 1 mL of Carrez reagent I (potassium ferrocyanide solution, c = 150 g L^−1^) and 1 mL of Carrez reagent II (zinc sulphate solution, c = 300 g L^−1^) were added to the sample and mixed. The volume of the flask was made up to the mark with water, mixed, and allowed to stand at ambient temperature for 20 min. Subsequently, the supernatant was filtered through a syringe membrane filter (0.22 µm) into a vial. The sample was analyzed by liquid chromatography on an Alliance 2695 instrument (Waters, Milford, MA, USA). Analysis conditions were as follows: mobile phase 100 % 20 mM phosphate buffer at pH = 2.7; Atlantis T3 Column—5 µm; 3.0 × 150 mm (Waters, Milford, MA, USA), column temperature 35 °C; flow rate 0.4 mL/min; UV detection on PAD 2996 (Waters, Milford, MA, USA) at 210 nm; and injection volume 10 µL. Data acquisition and evaluation were performed using Empower 2 software (Waters, Milford, MA, USA).

### 2.8. Validation Parameters

The detection limits (LODs) for each acid are the lowest points of the calibration lines and their values are listed in Table 1 (mg L^−1^). The repeatability of the method was determined by parallel analyses (n = 12) of an identical sample and is expressed as the coefficient of variation or relative standard deviation (RSD) for each acid in Table 2. Figure 1 shows chromatogram of organic acid standards.

### 2.9. Statistical Analysis

Data analysis was performed using Statistica 13.2 (StatSoft Inc., Tulsa, OK, USA). *p*-values less than 0.05 were considered statistically significant. One/two factor analysis of variance followed by Tukey’s HSD test was used for data analysis. In some cases (where the assumptions of ANOVA, i.e., normality of data and homogeneity of variances, were significantly violated), a non-parametric version of ANOVA, i.e., Kruskal–Wallis test followed by Dunn’s test with Bonferroni adjustment of *p*-values, was used (ceftiofur-containing samples—evaluation of pyruvic and succinic acid concentrations).

## 3. Results

The model experiment explored the effect of five cephalosporin antibiotics (cephalexin, cefoperazone, cefquinome, cefazolin, and ceftiofur) on the activity of five yoghurt cultures, expressed as changes in active and titratable acidity and through the determination of the concentration of the selected organic acids after fermentation. For all the cultures, a fermentation temperature of 43 °C was chosen and the fermentation process was terminated at a pH ≤ 4.6 of the blank. The profile of organic acids in fermented dairy products such as yoghurt is a useful indicator of the metabolic activity of starter cultures. The organic acids in the final products play an important role as natural preservatives, while at the same time contributing to the characteristic sensory properties of the products.

### 3.1. Active and Titratable Acidity

Figure 2 shows the pH value at the end of fermentation, and it is clear that ceftiofur showed the most significant inhibitory effect. In the ceftiofur-containing samples, the acidity of the samples at the end of fermentation fell only slightly (never falling below pH 6.0) from the value at the beginning of fermentation. The statistical analysis confirmed a highly statistically significant effect (*p* < 0.05) of ceftiofur on the pH value at the end of the fermentation process for all cultures. For instance, in the samples with ceftiofur inoculated with culture Z, the difference in the arithmetic means of the pH values at the beginning and end of fermentation was negligible (pH = 6.65 ± 0.01 and pH = 6.61 ± 0.03, respectively). The statistical analysis showed that the antibiotic was a statistically significant source of variability (*p* < 0.05; F-test), and that the culture did not statistically significantly affect the pH values. A post hoc test confirmed that only ceftiofur had a statistically significantly higher pH value than all the other cephalosporins and the blank (*p* < 0.05; Tukey’s HSD test).

Berruga et al. [9] focused on the effect of cephalosporins on the production of yoghurt from sheep’s milk when residues were present in milk due to the treatment of lactating ewes. In that study, the changes in pH values were monitored during the fermentation process in milk samples containing cephalexin and ceftiofur at concentrations of 50, 100, and 150 μg kg^−1^. Unlike ceftiofur, for cephalexin at levels ≥ MRL, no significant delay in the fermentation process was demonstrated compared to the control. Ceftiofur at 50 µg kg^−1^ (0.5 × MRL) delayed fermentation by 3.5 h compared to the control, and 100 µg kg^−1^ prolonged the delay by a further 40 min. An amount of 150 µg kg^−1^ caused a delay of more than 6 h. The results are consistent with our observation that ceftiofur showed a significant effect—practically from the beginning of incubation—on the activity of yoghurt cultures at the MRL level.

As can be seen from Figure 3, the arithmetic means of titratable acidity in the blank samples of all cultures at the end of fermentation ranged from 37.59 to 43.50 °SH, with the highest value measured for culture W (43.50 ± 0.59 °SH). The graph also shows clear differences in titratable acidity between the ceftiofur-containing samples and the other samples, including the blank controls. The ceftiofur-containing samples showed only a slight increase in titratable acidity during fermentation (a maximum of 0.1–0.6 °SH). Only one sample (containing ceftiofur and inoculated with culture W) showed a titratable acidity exceeding 10 °SH (13.30 ± 0.24 °SH).

The statistical analysis of the titratable acidity values of the cephalosporin-containing samples and the blanks at the end of fermentation showed that the antibiotic was a statistically significant source of variability (*p* < 0.05; F-test); thus, the culture did not statistically significantly affect the titratable acidity. A post hoc test confirmed that only ceftiofur was statistically significantly lower than all other cephalosporins and the blank with respect to titratable acidity (*p* < 0.05; Tukey’s HSD test). Novés et al. [10] showed that cephalexin present in milk at concentrations ≤ MRL inhibited the growth of *S. thermophilus* and caused changes in yoghurt acidity (pH and titratable acidity). In the present study, no statistically significant effect (*p* > 0.05) was observed on the titratable acidity at the end of fermentation for any culture in the samples containing cephalexin residues at MRL.

### 3.2. Changes in Lactic Acid Concentration

The data in Table 3 show that the main organic acid produced by the yoghurt cultures was lactic acid. The lactic acid concentration was quite low in the milk before fermentation (10.11 ± 5.39 mg 100 g^−1^) and ranged from 972.57 ± 87.22 to 1269.43 ± 60.48 mg 100 g^−1^ at the end of fermentation in the blank samples. A statistical evaluation confirmed the correlation between the observed lactic acid concentration and the cephalosporins in all the cultures (*p* < 0.05; F-test). A statistically significant effect (*p* < 0.01) of ceftiofur on lactic acid production was demonstrated for all the cultures. The lactic acid concentration at the end of fermentation was <50 mg 100 g^−1^ in the samples containing this antibiotic at a concentration of 1 MRL and inoculated with cultures V, X, Y, and Z, but in the sample inoculated with culture W the concentration was somewhat higher (216 mg 100 g^−1^). A statistically significant effect (*p* < 0.05) on lactic acid concentration was also found for the samples containing cefoperazone (culture Z), cefquinome (culture V), cefazolin (culture W), and cephalexin (cultures V, W, and X). For these cephalosporins, the differences in lactic acid concentration between the blank and enriched samples were not as significant (42–139 mg 100 g^−1^) as for the ceftiofur-containing samples (932–1056 mg 100 g^−1^). Table 3 lists the statistical significance of the differences in lactic acid concentration.

The main product of lactose catabolism by yoghurt culture is lactic acid, as confirmed by the high concentrations of this acid in our blank samples at the end of fermentation. If the growth of the yoghurt culture is inhibited, its lactic acid production is reduced. According to the literature, its concentration is highly variable in natural yoghurts, accounting for up to 75.7% of the organic acid content of yoghurt [22] and a concentration at the end of yoghurt production of 750 ± 74 mg 100 g^−1^. Fernandez-Garcia and McGregor [15] found 965 mg 100 g^−1^ lactic acid in natural yoghurt, while others [14] have found much lower concentrations (589 mg 100 g^−1^). Novés et al. [10] studied the effects of cephalexin at concentrations close to the MRL on the quality of yoghurt made from sheep milk. Their results showed that cephalexin concentrations ≤ MRL could inhibit the growth of mainly *Streptococcus thermophilus* and induce changes in the acidity (pH and titratable acidity) and the production mainly of the L(+)-—lactic acid isomer. The study showed that beta-lactam antibiotics, even at concentrations close to the MRL, can affect the fermentation of yoghurt, prolonging it and consequently changing the composition and texture of the product. Suhren [13] reported a significant inhibitory effect of ceftiofur at concentrations < MRL on the metabolism of yoghurt culture V2 with consequent changes in lactic acid production and pH. Our observations confirm the results of these studies [10,13], i.e., ceftiofur and cephalexin can inhibit lactic acid production at concentrations close to MRL.

### 3.3. Changes in Pyruvic Acid Concentration

Pyruvic acid is another organic acid produced by the metabolic activity of the yoghurt culture. If antibiotics inhibit the metabolism of the culture, the concentration of this acid is also affected. The inhibition of pyruvate production can be very significant because pyruvate is metabolized by decarboxylation to acetaldehyde, which is a key flavoring compound in yoghurt. The concentration of pyruvic acid in the samples at the beginning of fermentation was very low (arithmetic mean of 0.03 ± 0.09 mg 100 g^−1^), and the concentrations in the blank samples at the end of fermentation were 2.67–4.09 mg 100 g^−1^. For all the cultures, a statistical analysis confirmed the association between the observed pyruvic acid content and cephalosporins (*p* < 0.05; F-test). No statistically significant differences (*p* > 0.05) were found between the pyruvic acid levels in the samples containing cefoperazone residues and the blanks at the end of fermentation in any culture. In contrast, statistically significant differences (*p* < 0.05) were found between the samples containing the antibiotics ceftiofur and cephalexin and the blanks in all cultures. When cefquinome residues were present in the milk, a statistically significant effect was found only with culture Z, and with cultures W and Z in the presence of cefazolin residues.

The most significant effect on pyruvic acid formation and metabolism was observed in the ceftiofur-containing samples, where the pyruvic acid concentrations at the end of fermentation were close to (0.37 mg 100 g^−1^—culture W; 0.13 mg 100 g^−1^—culture Y) or below the detection limit (cultures V, X, and Z), indicating that no pyruvic acid was formed in these samples (Table 3). Similar to lactic acid, the results indicate a significant effect of ceftiofur on the fermentation process.

The increase in the pyruvic acid concentration in the blank samples at the end of fermentation confirmed that pyruvic acid is formed as an intermediate and/or initial metabolite of numerous biochemical reactions, in particular, carbohydrate, protein, and citric acid metabolism. Fernandez-Garcia and McGregor [15] reported that bacterial fermentation first results in the formation of pyruvic acid, which is subsequently converted to lactic acid and other metabolites with the help of a number of enzymes. The results showed a decrease in the pyruvic acid levels from the 6th hour of fermentation as well as subsequently during storage. Adhikari et al. [14] investigated the changes in the profiles of organic acids in plain set and stirred-type yogurts containing the starter culture and microencapsulated and nonencapsulated probiotic strains of either *Bifidobacterium longum B6* or *B. longum* ATCC 15708. In that study, both an increase and a decrease in the pyruvic acid content were observed during fermentation. The measured values at the end of fermentation were 3.2 mg 100 g^−1^ in the control samples not containing bifidobacteria. In the present study, similar concentrations were found in the blank samples.

Our results confirmed that pyruvic acid formation was greatly affected by ceftiofur and cephalexin residues. Cefazolin, cefquinome, and cefoperazone had a somewhat less significant effect.

### 3.4. Changes in Citric Acid Concentration

The arithmetic mean of the citric acid concentrations in the milk samples at the beginning of fermentation was 142.6 ± 24.62 mg 100 g^−1^. At the end of fermentation, the concentration in the blank samples ranged from 121.96 ± 16.32 to 145.44 ± 5.15 mg 100 g^−1^. For cultures V, W, X, and Z, we confirmed the association between the detected acid concentration and cephalosporins. The significance of the association was lowest in culture X and highest in culture Z. The citric acid concentrations in the samples containing cefazolin- and cefquinome-residues and in the blank samples showed no statistically significant differences (*p* > 0.05) after fermentation with the tested cultures (V, W, X, Y, and Z). As shown in Table 3, statistically significant differences (*p* < 0.05) were observed between the citric acid concentrations at the end of fermentation in the blank samples and in the samples containing cephalexin (culture W and Z), ceftiofur (culture V and Z), and cefoperazone (culture W).

The concentration of citric acid in the yoghurt should not be affected by the metabolic activity of conventional yoghurt cultures. Yoghurt starter culture does not have the ability to catabolize citric acid [23]. Citrate is a natural component of milk, formed in the mammary gland from precursors such as acetate and amino acids. Citric acid can be metabolized by some strains of the genera *Lactococcus*, *Leuconostoc*, and strains of probiotic bacteria, resulting in the formation of flavoring compounds (diacetyl and acetoin) in fermented products. Vénica et al. [22] concluded that the citric acid concentration does not change significantly during the production and storage of yoghurt. In natural yoghurts, without the addition of *β*-galactosidase, skimmed milk powder, whey protein concentrate, and sucrose, the citric acid concentration at the end of fermentation was 183 ± 3.82 mg 100 g^−1^—somewhat higher than in milk (176 ± 9.34 mg 100 g^−1^). After 14 days of storage, the concentration decreased to 169.57 ± 24.03 mg 100 g^−1^. However, Adhikari et al. [14] reported a decrease in the concentration of citric acid in yoghurt compared to that in milk before fermentation. The decrease was more significant in yoghurts containing bifidobacteria.

Our results suggest that an inhibition of the metabolic activity of the culture does not significantly reduce the concentration of citric acid. In the samples containing ceftiofur, the metabolic activity of all the cultures was almost completely inhibited, but a statistically significant effect (*p* < 0.05) on citric acid concentration was shown only for cultures V and Z.

### 3.5. Changes in Acetic Acid Concentration

The concentration of acetic acid in the milk before fermentation was low: 4.34 ± 1.25 mg 100 g^−1^. At the end of the fermentation process, the concentration was higher. In the blank samples, the concentration ranged from 9.21 ± 0.53 to 10.05 ± 0.90 mg 100 g^−1^. In all the cultures, the association between the detected acid concentration and cephalosporins was confirmed. The strength of the association was lowest in culture Z. Statistically significant differences (*p* < 0.05) in acetic acid concentrations in the blank and antibiotic-containing samples after the end of fermentation were observed only in the samples containing the antibiotic ceftiofur, and this was the case for all the cultures (Table 3).

Acetic acid is considered a volatile substance found in yoghurt. High acetic acid concentrations contribute to acidity and are generally associated with a ‘vinegary, tangy, or sour’ taste in the product. Higher concentrations of acetic acid are generally associated with heterofermentative glycolysis that is typical, for example, of bifidobacteria [24]. Low levels of acetic acid have been recorded during the fermentation and storage of traditional yoghurts [15,23]. The levels of acetic acid in milk fermented with basic yoghurt culture or with cultures containing *L. delbrueckii* ssp. *bulgaricus* are higher than in milk fermented with a culture containing only *S. thermophilus*. The concentration of acetic acid increases during storage [25].

The changes in acetic acid concentration confirm published conclusions that acetic acid concentration increases during fermentation and that this increase occurs even in yoghurts containing only homofermentative lactic acid bacteria (basal culture) [22]. Vénica et al. [22] also observed an increase in acetic acid content from 3.30 ± 0.14 mg 100 g^−1^ before fermentation to 7.87 ± 1.00 mg 100 g^−1^ at the end of yoghurt production in milk samples without the addition of milk powder and β-galactosidase. La Torre et al. [26] found out a significant increase in acetic acid concentration in yoghurts and fermented beverages containing probiotic strains of lactic acid bacteria. Acetic acid was also present in yoghurts containing traditional yoghurt culture, but only at low concentrations (7.6 mg 100 g^−1^). In the present study, higher acetic acid concentrations were observed in the blank samples after fermentation compering with those reported in the cited studies [22,26]. It is evident that significant changes in the concentration of this acid occurred only in the samples containing ceftiofur, which had a significant inhibitory effect on all the cultures tested.

### 3.6. Changes in the Concentration of Orotic Acid

The concentration of orotic acid in the milk before fermentation in our study was 4.53 ± 0.63 mg 100 g^−1^, and 3.7–4.88 mg 100 g^−1^ in the blank samples after fermentation. The association between the observed acid concentration and cephalosporins was not confirmed for culture X. The significance of the association was lowest in culture V. Statistically significant differences (*p* < 0.05) were observed between the concentrations of orotic acid at the end of fermentation in the blank samples and in the samples containing cefquinome (culture Y), ceftiofur (cultures W, Y, and Z), cephalexin (culture W), and cefazolin (cultures W and Z). No statistically significant differences (*p* > 0.05) were observed in the cefoperazone-containing samples in the concentrations of orotic acid in the blank and the cephalosporin-containing samples after fermentation (Table 4).

According to the literature data, the amount of orotic acid in milk depends on the breed, nutrition program, and stage of lactation [27]. Orotic acid is an intermediate in nucleotide synthesis and is known to be an important growth factor for yoghurt cultures. Its concentration decreases during yoghurt production and storage. The concentration in milk was 87 ± 2.0 mg 100 g^−1^ dry matter and in yoghurt was 76.1 ± 1.1 mg 100 g^−1^ dry matter [16]. Saidi and Warthesen [28] reported 69–74 mg L^−1^ orotic acid in cow’s milk. That study confirmed that the content of orotic acid decreases by up to 45% during fermentation.

The concentration of orotic acid in milk before and after fermentation in the blank samples in the present study was lower than the values reported in the cited studies. The most significant effect with respect to changes in orotic acid concentrations was observed in the samples containing ceftiofur residues.

### 3.7. Changes in Uric Acid Concentration

In our study, uric acid was present in low concentrations (1.22 ± 0.50 mg 100 g^−1^) at the beginning of fermentation, while in the blank samples it ranged between 1.15 and 1.60 mg 100 g^−1^. The association between the observed concentration and the cephalosporins was confirmed in all cultures. The significance of the association was highest in culture V and lowest in culture X (Table 4). Statistically significant differences (*p* < 0.05) were observed between the uric acid concentrations at the end of fermentation in the blank sample and in the sample containing cefoperazone (cultures V, W, and Z), cefquinome (cultures V and Y), ceftiofur (cultures X and Y), cephalexin (cultures W and Z), and cefazolin (cultures W, X, and Z).

Uric acid occurs naturally in milk as an intermediate of nitrogen metabolism at concentrations of 5–8 mg kg^−1^ [23]. Data on uric acid concentrations in yoghurt vary in the literature. According to a study by Fernandez-Garcia and McGregor [15], uric acid concentrations showed only a slight decrease (from 34.7 µg g^−1^ in milk to 32.7 µg g^−1^ in yoghurt). On the other hand, Adhikari et al. [14] concluded that the concentration of this acid increased 2.5–3 times during fermentation, irrespective of the type of yoghurt or the bifidobacterial content. The results of the cited study indicated that the increase in uric acid concentration is the result of the activity of the yoghurt culture, reporting concentrations of 3.2 mg 100 g^−1^ (set yoghurt) and 3.0 mg 100 g^−1^ (stirred yoghurt) in yoghurt. Guler et al. [29] found a statistically significant decrease in uric acid content in a fermented dairy product containing a yoghurt culture with the addition of kefir grains during storage. The results of that study reported the significant decrease in concentrations of uric acid due to the utilization of this acid by lactic acid bacteria using it as a source of nitrogen to form carbon dioxide and ammonia.

The uric acid levels in our blank samples were lower, but a slight increase in concentration was observed at the end of fermentation, which is in agreement with that seen by Adhikari et al. [14].

### 3.8. Changes in Formic Acid Concentration

We measured 16.93 ± 2.21 mg 100 g^−1^ of formic acid in milk before fermentation, and 15.62–17.6 mg 100 g^−1^ at the end of fermentation. The association between acid concentration and cephalosporins was not demonstrated for cultures V and Y (Table 3). There were no statistically significant differences (*p* > 0.05) in the formic acid concentrations at the end of fermentation compared to the blank for samples containing cefoperazone and cefquinome residues in all cultures. Statistically significant differences (*p* < 0.05) in the formic acid content were found only in samples containing ceftiofur (culture W), cephalexin (culture W), or cefazolin (culture X).

In yoghurt culture, streptococci promote the growth of lactobacilli by releasing formic acid from pyruvic acid under anaerobic conditions and rapidly producing CO_2_ (via a pyruvate–formate lyase reaction). This reaction is essential for the anaerobic conversion of pyruvate to formate and acetyl-CoA. Acetyl-CoA is further metabolized to form ethanol and acetate. In pasteurized milk, the formate content may also be affected by the intensity of the heat treatment (formic acid is formed as a product of the Maillard reaction due to the decomposition of hydroxymethylfurfural [24]).

Both increases and decreases in this organic acid have been seen to occur in yoghurt, and it is not possible to directly compare the results of our study with the literature. Gonzáles de Llano et al. [30] determined the formic acid content in milk (41.05 ± 1.02 mg 100 g^−1^) and in milk fermented with different starter cultures, but not with yoghurt culture. Other studies focusing on the determination of organic acids in fermented dairy products [14,16,23] did not measure the formic acid content in yoghurt. Although Fernandez-Garcia and McGregor [15] determined the formic acid content in yoghurt (717.7 µg g^−1^), their results included an unknown peak in the chromatogram.

### 3.9. Changes in Oxalic Acid Concentration

Oxalic acid is present in yoghurts and, together with other non-volatile (pyruvic and succinic) and volatile (formic, acetoacetic, and propionic) acids, as well as other components (acetaldehyde and acetone), which contribute to the aroma of yoghurts [31]. We observed that 8.11 ± 3.36 mg 100 g^−1^ oxalic acid was present in milk before fermentation. After fermentation, a relatively wide range of concentrations was found in the blank samples (6.12–14.34 mg 100 g^−1^). In all the cultures, the association between the measured acid concentration and cephalosporins was confirmed. The significance of the association was highest for culture Z and lowest for culture W (Table 4). There were no statistically significant differences (*p* > 0.05) in the oxalic acid concentrations compared to the blank for the cefoperazone-containing samples in any culture. We found statistically significant differences (*p* < 0.05) in oxalic acid content between the blank samples and those that contained cefquinome (cultures V and Z), ceftiofur (cultures V, Y, and Z), cephalexin (cultures X, Y, and Z) or cefazolin residues (cultures W and Y; Table 4). The oxalic acid concentrations in the blank samples at the end of fermentation were lower than those published by Tormo and Izco [16], who reported 73.0 ± 22.7 mg 100 g^−1^ of oxalic acid in commercially produced yoghurt. In our study, the oxalic acid concentrations in the blank samples at the end of fermentation were lower.

### 3.10. Changes in Succinic Acid Concentration

The concentration of succinic acid in our experiments was very low (1.43 ± 0.54 mg 100 g^−1^) at the beginning of fermentation. At the end of fermentation, the concentrations in the blank samples were higher, ranging from 1.84–7.69 mg 100 g^−1^. This increase in the succinic acid concentration during fermentation is consistent with the literature. We also confirmed the association between the measured succinate content and cephalosporins for all cultures. The significance of the association was lowest for culture V and highest for culture Z. There were no statistically significant differences (*p* > 0.05) in the succinic acid content compared to the blank in the samples with added cefoperazone and cefquinome in allcultures used. In the samples containing ceftiofur, statistically significant differences compared to the blank (*p* < 0.05) were demonstrated for three cultures (W, Y, and Z), for culture X in the samples containing cephalexin, and for culture W in the samples containing cefazolin (Table 4).

Among the metabolites that are important for the symbiotic relationship between the two strains of yoghurt culture, fumaric acid (produced by the activity of *S. thermophilus*) is metabolized by *L. bulgaricus* to form succinic acid [32]. Succinic acid and other non-volatile (pyruvic and oxalic) acids contribute to the typical aroma of fermented dairy products [33].

No information on the concentration of this organic acid in yoghurt is available in the literature, although numerous studies have focused on the determination of organic acids in dairy products and yoghurt [14,15,16,23,30,33,34,35].

### 3.11. Principal Component Analysis

The results of the principal component analysis confirmed that ceftiofur had the greatest effect on all the observed parameters. As can be seen in Figure 4, only samples with ceftiofur (left) were clearly separated (in the direction of the horizontal axis).

The significant effect of ceftiofur on the activity of the cultures is also seen in Figure 5, which shows that there is a relatively clear separation (in the direction of the vertical axis) between the samples with culture W (bottom right) and some samples with culture Z (top left) or culture Y (bottom left), which correspond to the samples containing ceftiofur.

Table 5 reveals that the horizontal PC1 axis was strongly positively correlated with titratable acidity (SH) values, pyruvic acid, and lactic acid concentrations; the samples with significant changes of these parameters are clustered on the right. This axis, in turn, is strongly negatively correlated with pH, so that the samples with high pH are clustered to the left. This table, in conjunction with Figure 4, clearly indicates that the ceftiofur samples had low titratable acidity (SH), pyruvic, and lactic acid levels as well as high pH values compared to all other samples, regardless of the culture used.

For the vertical PC2 axis, the values of the correlation coefficients were not as pronounced, but more pronounced positive correlations were observed for oxalic, orotic acid, and uric acids (the samples near the top were those with higher values). The more pronounced negative correlations were seen for succinic acid, so the samples with a higher succinic acid content tended to be near the bottom. In conjunction with Figure 5, it is clear that the samples with culture W (without ceftiofur) had higher levels of succinic acid and lower levels of oxalic, orotic, and uric acids compared to all other samples.

## 4. Conclusions

The aim of the study was to determine the sensitivity of yoghurt cultures to residual levels of selected cephalosporin antibiotics (cephalexin, cefoperazone, cefquinome, cefazolin, and ceftiofur) at a concentration of 1 MRL in milk. Five thermophilic yoghurt cultures were selected containing *Lactobacillus delbrueckii* subsp. *bulgaricus* and *Streptococcus thermophilus*, strains typical for yoghurt culture. The activity of the cultures was evaluated by monitoring the active and titratable acidity and by measuring the concentration of organic acids (lactic, pyruvic, citric, acetic, orotic, oxalic, formic, uric, and succinic) at the end of fermentation. Among the tested cephalosporins, ceftiofur had the greatest effect on the activity of the yoghurt cultures, with a statistically highly significant effect (*p* < 0.05) on the pH, titratable acidity, and on the contents of lactic, pyruvic, and acetic acids. Ceftiofur caused the complete inhibition of the metabolic activity of yoghurt cultures at the MRL concentration. The pH values, titratable acidity, and especially the organic acid (lactic and pyruvic acid) concentrations are the most important parameters that reflect the inhibition of metabolic activity in the yoghurt culture. This inhibitory effect on the metabolism of yoghurt cultures was further demonstrated by evaluating the concentrations of organic acids (lactic and pyruvic acids) in the samples containing other cephalosporins. The conclusion, therefore, is that even very low concentrations of antimicrobial residues in milk (e.g., 100 µg kg^−1^ = MRL for ceftiofur) can significantly inhibit the growth and activity of dairy cultures. The MRLs set by legislation for antimicrobials are of undisputed importance for the safety of raw materials and foods of animal origin but are not sufficient to always ensure the technological safety of the raw material.

This study has some limitations. It is necessary to mention some circumstances that must be considered: The present study demonstrated the negative influence of cephalosporins on the metabolic activity of yoghurt cultures during the fermentation of spiked milk. In the naturally contaminated milk, the antibiotics can be present in milk in different forms. “Total residues” of drug in the edible products of the treated animals include not only the parent drug but also many other products of drug metabolism (free metabolites of the parent drug, conjugates to small molecules and macromolecules, covalently bound metabolites, and drug fragments). Not all metabolites are bioactive. The isolation and identification of all these metabolites are difficult [6]. Di Rocco et al. [36] drew attention to the fact that the differences between spiked and naturally contaminated milk should be considered when performing such investigations. Pasteurized milk was used as a medium for the preparation of artificially fortified samples. In this study, the antibiotics were added into the milk after a heat treatment. Temperature is one of the most important factors since it is responsible for the inactivation of the drugs. Roca et al. [37] showed that heat treatments at high temperatures and long times (e.g., 120 °C for 20 min) led to a further degradation of β-lactam antibiotics with percentages close to 100% for cefoperazone and cefuroxime. In contrast, when milk was subjected to heat treatments at lower temperatures and times (e.g., 72 °C for 15 s), the degradation of β-lactam antibiotics in milk did not exceed 1% for the 10 antibiotics tested. Nowadays, new directions of research are focused on the migration of drug residues in dairy derivates [36,38,39] and their degradation during manufacture [40]. For example, the research published by Grunwald and Petz [40] demonstrated that the fermentation process was responsible for decreasing penicillin residues levels. However, there are many factors that could affect the concentration of residual drugs during the manufacture of yoghurt (e.g., the heat treatment of the milk, the fermentation temperature and time, the binding of residues to milk proteins etc.).

## Figures and Tables

**Figure 1 foods-11-02751-f001:**
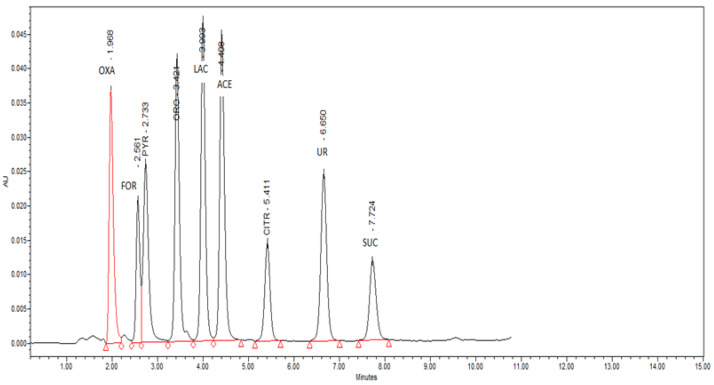
Chromatogram of organic acid standards—OXA—oxalic acid; FOR—formic acid; PYR—pyruvic acid; ORO—orotic acid; LAC—lactic acid; ACE—acetic acid; CITR—citric acid; UR—uric acid; SUC—succinic acid; AU—absorbance units.

**Figure 2 foods-11-02751-f002:**
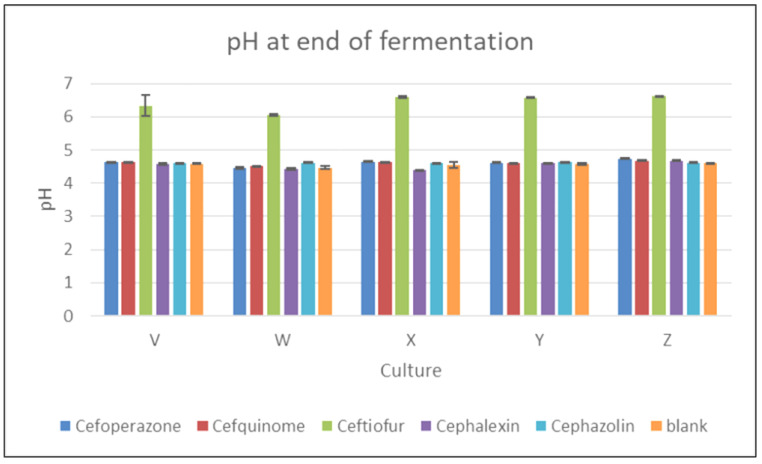
The effect of five cephalosporin antibiotics (cephalexin, cefoperazone, cefquinome, cefazolin, and ceftiofur) on the activity of five yoghurt cultures (V, W, X, Y, and Z) in relation to the pH value at the end of fermentation.

**Figure 3 foods-11-02751-f003:**
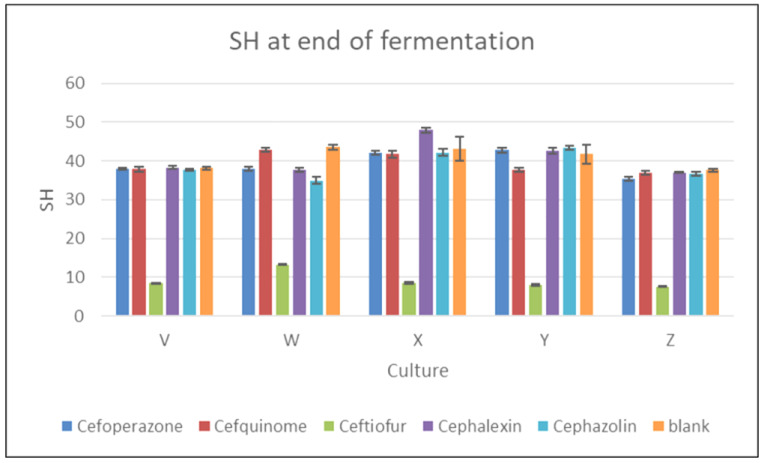
The effect of five cephalosporin antibiotics (cephalexin, cefoperazone, cefquinome, cefazolin, ceftiofur) on the activity of five yoghurt cultures (V, W, X, Y, and Z) in relation to titratable acidity (°SH) at the end of fermentation.

**Figure 4 foods-11-02751-f004:**
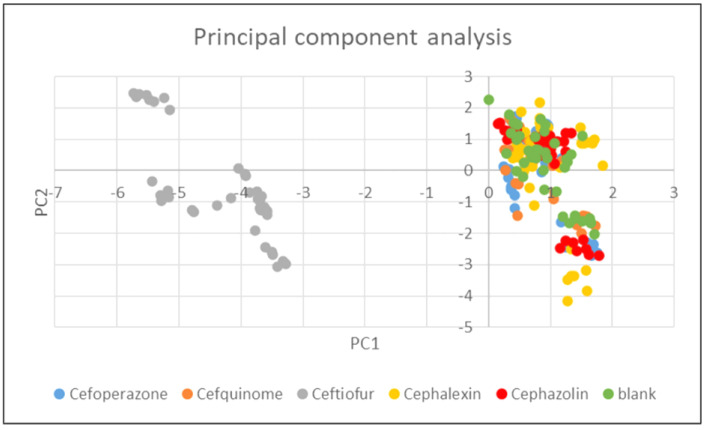
Principal Component Analysis: Effect of cephalosporins on pH value, titratable acidity value, and organic acids concentration at the end of fermentation.

**Figure 5 foods-11-02751-f005:**
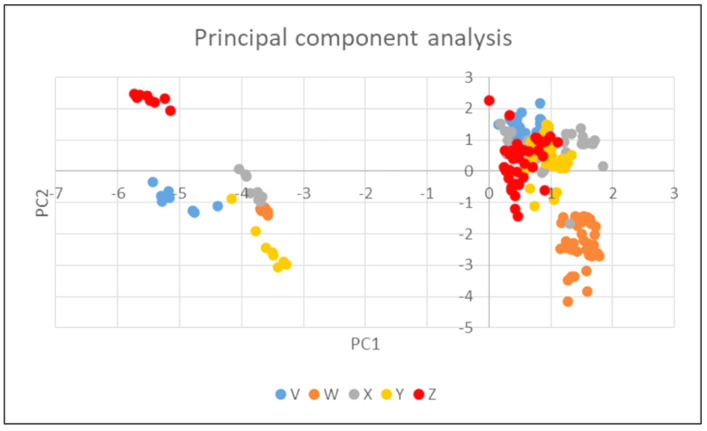
Principal Component Analysis: Effect of cultures (V, W, X, Y, and Z) on pH value, titratable acidity value, and organic acids concentration at the end of fermentation.

**Table 1 foods-11-02751-t001:** Calibration lines for organic acids.

Acid	LOD (mg L^−1^)	Linear Range (mg L^−1^)	R	Calibration Line Equation
OXA	0.20	0.20–32.7	0.999757	Y = 2.23 × 10^4^ + 2.19 × 10^3^
FOR	0.18	0.18–90.0	0.998717	Y = 1.14 × 10 ^4^ + 4.09 × 10^3^
PYR	0.10	0.10–60.0	0.999618	Y = 1.12 × 10^4^ + 6.41 × 10^4^
ORO	0.03	0.03–2.8	0.999942	Y = 1.83 × 10^5^ − 3.47 × 10^3^
LAC	9.70	9.7–1446.7	0.999949	Y = 6.63 × 10^2^ − 3.52 × 10^3^
ACE	8.25	8.2–1256.4	0.999794	Y = 8.09 × 10^2^ − 6.57 × 10^3^
CITR	1.10	1.1–236.6	0.999831	Y = 1.52 × 10^3^ − 1.34 × 10^3^
UR	0.08	0.08–14.5	0.999556	Y = 4.67 × 10^4^ + 2.70 × 10^3^
SUC	3.06	3.06–459.8	0.999789	Y = 8.06 × 10^2^ − 1.32 × 10^3^

R—linearity coefficient; LOD—limit of detection, OXA—oxalic acid; FOR—formic acid; PYR—pyruvic acid; ORO—orotic acid; LAC—lactic acid; ACE—acetic acid; CITR—citric acid; UR—uric acid; SUC—succinic acid.

**Table 2 foods-11-02751-t002:** Validation parameters of the RP HPLC method for organic acids (mg L^−1^).

	OXA	FOR	PYR	ORO	LAC	ACE	CITR	UR	SUC
Mean	23.29	25.51	250.28	0.69	420.33	387.45	75.92	4.40	145.34
SD	1.44	0.77	3.39	0.04	9.29	8.219	2.43	0.15	2.79
RSD	6.17	3.00	1.35	5.07	2.21	2.12	3.20	3.43	1.92

SD—standard deviation, RSD = coefficient of variation; mean—arithmetic mean; OXA—oxalic acid; FOR—formic acid; PYR—pyruvic acid; ORO—orotic acid; LAC—lactic acid; ACE—acetic acid; CITR—citric acid; UR—uric acid; SUC—succinic acid.

**Table 3 foods-11-02751-t003:** Concentration of organic acids at the end of fermentation in samples fortified with cephalosporins and in blank samples (mg 100 g^−1^).

Culture	Blank Sample	Cefoperazon	Cefquinome	Cefazolin	Ceftiofur	Cefalexin	Eta
Lactic acid							
V	1104.96 ± 32.94 ^a^	1092.66 ± 25.46 ^ae^	1030.16 ± 19.18 ^bc^	1121.82 ± 21.29 ^a^	48.32 ± 3.47 ^d^	1062.23 ± 42.37 ^ce^	0.9952
W	1269.43 ± 60.48 ^bc^	1269.88 ± 11.96 ^cd^	1305.18 ± 33.30 ^bd^	1090.99 ± 55.90 ^a^	216.11 ± 5.83 ^e^	1089.88 ± 164.58 ^a^	0.9603
X	980.06 ± 94.66 ^a^	951.78 ± 23.45 ^a^	941.55 ± 44.67 ^a^	933.75 ± 26.66 ^a^	25.88 ± 1.34 ^b^	1164.75 ± 22.09 ^c^	0.9847
Y	972.57 ± 87.22 ^a^	924.06 ± 50.32 ^a^	889.77 ± 87.66 ^a^	943.66 ± 32.20 ^a^	40.41 ± 6.42 ^b^	962.07 ± 86.97 ^a^	0.9623
Z	1058.58 ± 71.37 ^b^	919.90 ± 69.97 ^a^	1011.08 ± 93.34 ^b^	1077.27 ± 33.63 ^b^	26.82 ± 0.68 ^c^	1069.51 ± 38.00 ^b^	0.9756
	Pyruvic acid						
V	3.42 ± 0.82 ^a^	3.31 ± 0.35 ^a^	3.27 ± 0.16 ^a^	2.84 ± 0.22 ^a^	ND	4.59 ± 0.29 ^b^	0.9259
W	2.67 ± 0.30 ^a^	2.95 ± 0.52 ^a^	2.83 ± 0.30 ^a^	3.78 ± 0.42 ^d^	0.38 ± 0.08 ^b^	1.97 ± 0.30 ^c^	0.9033
X	3.56 ± 0.60 ^a^	3.64 ± 0.12 ^a^	3.37 ± 0.25 ^a^	4.13 ± 1.18 ^ab^	ND	4.55 ± 0.19 ^b^	0.8765
Y	4.09 ± 0.53 ^a^	4.09 ± 0.25 ^a^	4.48 ± 0.50 ^a^	3.90 ± 0.15 ^a^	0.13 ± 0.06 ^b^	3.07 ± 0.70 ^c^	0.9216
Z	3.19 ± 0.29 ^a^	2.87 ± 0.15 ^a^	2.66 ± 0.30 ^a^	4.46 ± 0.32 ^b^	ND	4.60 ± 0.17 ^b^	0.9768
	Acetic acid						
V	9.21 ± 0.53 ^a^	9.94 ± 0.50 ^a^	9.15 ± 0.33 ^a^	8.87 ± 0.73 ^a^	42.86 ± 3.87 ^b^	9.05 ± 0.56 ^a^	0.9829
W	10.05 ± 0.90 ^ac^	9.55 ± 0.80 ^ac^	10.84 ± 1.17 ^a^	8.75 ± 0.37 ^c^	24.16 ± 1.37 ^b^	10.37 ± 0.77 ^a^	0.9694
X	9.59 ± 0.67 ^a^	10.03 ± 1.15 ^a^	9.41 ± 0.71 ^a^	9.74 ± 0.57 ^a^	4.40 ± 0.39 ^b^	10.27 ± 0.64 ^a^	0.8875
Y	9.62 ± 0.46 ^ac^	9.44 ± 0.42 ^a^	9.54 ± 0.94 ^ac^	9.49 ± 0.28 ^ac^	5.22 ± 0.86 ^b^	10.55 ± 0.99 ^c^	0.8520
Z	9.67 ± 0.56 ^a^	9.25 ± 0.28 ^a^	10.28 ± 1.34 ^a^	10.38 ± 0.36 ^a^	6.54 ± 4.05 ^b^	10.55 ± 0.30 ^a^	0.3762
	Formic acid						
V	17.60 ± 1.16 ^a^	17.67 ± 0.86 ^a^	16.82 ± 0.85 ^a^	17.98 ± 0,52 ^a^	16.92 ± 1.39 ^a^	17.95 ± 0.70 ^a^	0.1878
W	17.06 ± 1.59 ^a^	17.53 ± 0.33 ^a^	18.02 ± 1.03 ^a^	17.80 ± 0.58 ^a^	20.20 ± 0.18 ^b^	15.11 ± 1.67 ^c^	0.6629
X	16.32 ± 3.06 ^a^	16.25 ± 0.92 ^a^	15.76 ± 0.51 ^a^	20.43 ± 4.47 ^b^	16.19 ± 1.84 ^a^	17.58 ± 0.41 ^ab^	0.3091
Y	15.62 ± 1.84 ^a^	16.54 ± 1.03 ^a^	15.01 ± 1.58 ^a^	15.86 ± 0.50 ^a^	15.67 ± 2.10 ^a^	16.17 ± 1.41 ^a^	0.0918
Z	17.04 ± 3.84 ^ab^	14.79 ± 1.02 ^a^	15.45 ± 1.22 ^a^	15.41 ± 0.26 ^a^	18.51 ± 0.71 ^b^	18.63 ± 0.88 ^b^	0.4285
	Citric acid						
V	145.03 ± 6.67 ^ac^	142.91 ± 4.40 ^a^	147.95 ± 2.98 ^ac^	143.22 ± 1.62 ^a^	157.08 ± 8.91 ^b^	151.90 ± 3.52 ^bc^	0.4812
W	145.45 ± 5.15 ^bc^	131.10 ± 7.84 ^a^	145.12 ± 5.44 ^bc^	139.12 ± 4.76 ^ac^	151.51 ± 1.71 ^b^	118.07 ± 10.88 ^d^	0.7373
X	135.72 ± 15.57 ^ab^	137.96 ± 3.26 ^ab^	139.40 ± 5.53 ^b^	134.70 ± 5.94 ^ab^	127.15 ± 7.04 ^a^	144.45 ± 2.33 ^b^	0.3064
Y	121.96 ± 16.32 ^a^	122.89 ± 8.95 ^a^	125.52 ± 11.76 ^a^	125.20 ± 2 0.43 ^a^	123.14 ± 16.73 ^a^	127.56 ± 6.81 ^a^	0.0258
Z	144.04 ± 16.30 ^a^	134.35 ± 9.47 ^a^	137.37 ± 12.70 ^a^	142.66 ± 4.26 ^a^	153.08 ± 11.87 ^b^	174.70 ± 6.42 ^c^	0.9360

Values within each row with different superscripts are significantly different (*p* < 0.05). Data are presented as the mean ± standard deviation; V, W, X, Y, and Z—type of yoghurt culture added; ND—not detected; Eta—effect size.

**Table 4 foods-11-02751-t004:** Concentration of organic acids at the end of fermentation in samples fortified with cephalosporins and in blank samples (mg 100 g^−1^).

Culture	Blank Sample	Cefoperazon	Cefquinome	Cefazolin	Ceftiofur	Cefalexin	Eta
Uric acid							
V	1.46 ± 0.11 ^c^	1.71 ± 0.06 ^a^	2.25 ± 0.06 ^b^	1.49 ± 0.04 ^c^	1.39 ± 0.12 ^c^	1.53 ± 0.17 ^c^	0.8879
W	1.15 ± 0.18 ^c^	1.38 ± 0.15 ^a^	1.23 ± 0.15 ^bc^	0.77 ± 0.14 ^d^	1.03 ± 0.06 ^c^	0.58 ± 0.23 ^ad^	0.7980
X	1.42 ± 0.22 ^a^	1.52 ± 0.19 ^ac^	1.53 ± 0.14 ^ac^	1.71 ± 0.08 ^cb^	1.80 ± 0.09 ^b^	1.32 ± 0.16 ^a^	0.5289
Y	1.22 ± 0.28 ^ad^	1.49 ± 0.12 ^ab^	1.65 ± 0.27 ^b^	1.15 ± 0.13 ^d^	0.40 ± 0.05 ^c^	1.07 ± 0.25 ^d^	0.7900
Z	1.60 ± 0.13 ^a^	1.22 ± 0.37 ^b^	1.55 ± 0.40 ^ab^	1.76 ± 0.04 ^a^	1.69 ± 0.04 ^a^	0.01 ± 0.01 ^c^	0.8735
Oxalic acid							
V	14.34 ± 0.67 ^a^	13.43 ± 0.81 ^a^	11.36 ± 1.89 ^bc^	14.14 ± 0.98 ^a^	10.69 ± 0.68 ^c^	14.68 ± 1.07 ^a^	0.6598
W	6.12 ± 0.63 ^abc^	7.10 ± 1.21 ^ad^	5.80 ± 0.28 ^b^	7.53 ± 1.10 ^d^	5.73 ± 0.25 ^bc^	5.96 ± 1.06 ^ab^	0.3953
X	6.04 ± 1.73 ^acd^	6.04 ± 1.73 ^ad^	4.85 ± 0.48 ^ac^	7.37 ± 0.74 ^bd^	4.59 ± 0.41 ^c^	8.85 ± 2.28 ^b^	0.5743
Y	9.50 ± 1.61 ^ad^	9.50 ± 1.60 ^ac^	9.20 ± 2.43 ^d^	12.74 ± 2.78 ^c^	6.12 ± 1.32 ^b^	5.52 ± 0.96 ^b^	0.6930
Z	13.52 ± 3.74 ^a^	13.52 ± 3.74 ^ab^	10.91 ± 0.83 ^b^	12.53 ± 0.51 ^ab^	22.08 ± 1.24 ^c^	19.52 ± 0.50 ^d^	0.8636
Succinic acid							
V	2.03 ± 0.28 ^a^	1.82 ± 010 ^a^	1.77 ± 0.08 ^a^	1.81 ± 0.09 ^a^	1.95 ± 0.22 ^a^	1.79 ± 0.13 ^a^	0.2359
W	7.69 ± 1.26 ^a^	8.23 ± 0.83 ^a^	8.41 ± 0.71 ^ac^	9.94 ± 1.32 ^c^	4.17 ± 0.39 ^b^	7.72 ± 1.32 ^a^	0.7398
X	1.89 ± 0.20 ^a^	1.89 ± 0.21 ^a^	1.75 ± 0.07 ^a^	1.99 ± 0.27 ^a^	1.44 ± 0.27 ^a^	3.20 ± 1.25 ^b^	0.5075
Y	1.85 ± 0.08 ^a^	1.81 ± 0.09 ^a^	1.91 ± 0.19 ^a^	1.96 ± 0.20 ^a^	0.86 ± 0.09 ^b^	1.81 ± 0.11 ^a^	0.8838
Z	1.95 ± 0.14 ^ab^	1.82 ± 0.07 ^a^	1.84 ± 0.10 ^ab^	2.05 ± 0.26 ^b^	ND	1.91 ± 0.15 ^ab^	0.9610
Orotic acid							
V	4.77 ± 0.16 ^ab^	4.83 ± 0.43 ^a^	4.45 ± 0.07 ^b^	4.85 ± 0.04 ^a^	4.62 ± 0.21 ^ab^	4.63 ± 0.12 ^ab^	0.2951
W	3.70 ± 0.10 ^a^	3.79 ± 0.37 ^a^	3.71 ± 0.02 ^a^	3.27 ± 0.06 ^d^	4.91 ± 0.04 ^b^	2.73 ± 0.41 ^c^	0.8897
X	4.88 ± 0.64 ^a^	5.10 ± 0.24 ^a^	5.28 ± 0.19 ^a^	4.90 ± 0.14 ^a^	5.15 ± 0.30 ^a^	4.93 ± 0.09 ^a^	0.1754
Y	4.88 ± 0.52 ^ad^	4.78 ± 0.26 ^a^	4.09 ± 0.39 ^bc^	4.88 ± 0.16 ^a^	3.85 ± 0.48 ^c^	5.43 ± 0.36 ^d^	0.6583
Z	3.87 ± 0.29 ^ac^	3.73 ± 0.29 ^ac^	3.91 ± 0.38 ^abc^	4.29 ± 0.13 ^d^	4.27 ± 0.09 ^bd^	4.01 ± 0.13 ^bcd^	0.4183

Values within each row with different superscripts are significantly different (*p* < 0.05). Data are presented as the mean ± standard deviation; V, W, X, Y, and Z—type of yoghurt cultures added; ND–not detected; Eta–effect size.

**Table 5 foods-11-02751-t005:** Correlations of each parameter with the two principal components, i.e., the horizontal (PC1) and vertical axis (PC2).

Parameter	PC1	PC2
Ph	−0.97	−0.12
titratable acidity	0.96	0.19
oxalic acid concentration	−0.15	0.61
formic acid concentration	−0.15	0.24
pyruvic acid concentration	0.85	0.35
orotic acid concentration	−0.15	0.63
lactic acid concentration	0.95	0.08
acetic acid concentration	−0.41	−0.15
citric acid concentration	−0.48	0.42
uric acid concentration	−0.08	0.64
succinic acid concentration	0.37	−0.69

## Data Availability

Data is available upon request to the authors.

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
