# Peer review of "Effect of Cephalosporin Antibiotics on the Activity of Yoghurt Cultures"

_foods, 2022, doi:10.3390/foods11182751_

Round 1

Reviewer 1 Report

This was an excellent report on very thorough research. I have a few substantive questions/comments to consider:

- There was no mention of the limitations of the study. Also, mention how your results compare to another study, which was published very recently.

- Academic writing should be objective. The language of academic writing should therefore be impersonal, and should not include personal pronouns, emotional language or informal speech. Use of personal pronouns (I/ my/ our/ us/ etc) can make the tone of writing too subjective. ( see the lines  57,81,206,237,240,267,313, 339, 342, 343,407, 435,459,463,508,578 and 582)

- what does the symble (Eta)  mean ? see ( line 325)

- Sometimes the authors write Fig and other place wrote figure ( see the line 252)

- Chromatograms of HPLC need to be inserted to the main text.

-Conclusion needs to be revised, and pay more attention on innovations in this research

- New references are needed to show the recent works either in introduction section or discussion section 

Author Response

Author response (reviewer 2)

Dear reviewer,

Our team of authors wish to express our thanks for taking the time to conduct a review of our manuscript entitled "Effect of cephalosporin antibiotics on the activity of yoghurt cultures", as well as for your insightful comments, questions and recommendations. Following your suggestions, we have corrected the manuscript and highlighted the changes in MS Word using the Track Changes function.

Attached answers to individual comments:

Comments 1:

There was no mention of the limitations of the study. Also, mention how your results compare to another study, which was published very recently.

Amendment:

The results are discussed with the scientific literature that was related to the topic of the study. Revised conclusion includes limitations of the study (lines 607-633).

Unfortunately, we probably skip the study and we are afraid that we cannot find it now, anyway. Could you please advise us which one you meant?

Comments 2:

Academic writing should be objective. The language of academic writing should therefore be impersonal, and should not include personal pronouns, emotional language or informal speech. Use of personal pronouns (I/ my/ our/ us/ etc) can make the tone of writing too subjective. (see the lines 57,81,206,237,240,267,313, 339, 342, 343,407, 435,459,463,508,578 and 582)

Amendment:

The changes were made and are highlighted in the text (lines 57; 80-81; 212; 243; 247; 273-274; 320-321; 348-349; 351-358; 419-420; 422;450; 453; 471; 473-474; 524; 594; 597-598).

Comments 3:

What does the symble (Eta) mean? see (line 325)

Amendment:

Eta-squared (Eta= effect size) is a descriptive measure of the strength of association between independent and dependent variables in the sample. Eta squared is a measure of effect size that is commonly used in ANOVA models. Explanation of this unit was added (line 335, 399).

Comments 4:

Sometimes the authors write Fig and other place wrote figure (see the line 252)

Amendment:

The abbreviation fig was replaced by the word figure (line 229)

Comments 5:

Chromatograms of HPLC need to be inserted to the main text.

Amendment

Since it is not possible to present the chromatograms of individual samples due to the large number of measured samples. The chromatogram of the standards measurement was inserted into the methodology section (lines 177-178, 197-201).

Comment 6:

Conclusion needs to be revised, and pay more attention on innovations in this research

Amendment:

Conclusion was revised (lines 586-633).

Comments 7:

New references are needed to show the recent works either in introduction section or discussion section 

Amendment

New references were added to the chapter references (lines 723-732).

We hope that the modifications we made satisfy your requirements.

Yours sincerely

Pavlina Navratilova

Reviewer 2 Report

The aim of our study was to determine the effect on the activity of yoghurt cultures of residues of selected cephalosporin antibiotics. From the point of view of the production of this type of food, it is an extremely important and interesting topic. The presence of antibiotics and other undesirable substances in the raw material has a decisive influence on the technological process. In this case, we are talking about lactic acid fermentation with the use of microorganisms. The presence and concentration of antibiotics is regulated by law. Legislation may vary from country to country, but small residues of undesirable substances can cause great financial and health damage. The work shows a relatively wide spectrum of research, because 5 lactic acid fermentation bacteria cultures and five types of antibiotics were used. In addition, a number of studies were carried out, giving results interesting from the point of view of the research question asked. The work is written in a clear and easy to understand manner. With its topic, it draws attention to the possible contamination of raw materials for food production with antibiotics, and thus possible technological problems. The work causes the reader to reflect more deeply on the problem. Because if the lactic acid bacteria do not work with such low concentrations of the antibiotic, what can happen to the microbiota and what impact it will have on human health. Going further, let's assume that the milk has not been fermented, but transferred for further processing for sale as a finished product, e.g. UHT milk. Technologically, the presence of antibiotics does not interfere with this process. The legislation of a given country may accept such low concentrations of the antibiotic in milk. At this point, we have a product that is harmful to health - it will inhibit the activity of lactic acid bacteria in the digestive tract. Hence my recommendation to accept the paper for publication. Interesting research supported by numerous results with a topic that touches not only technological but also health problems. Of course, the authors of the research in this area did not carry out, because it was not the subject of their work, nevertheless, the work affects the reader on two levels. The conclusions are correct and related to the topic of the work, they also refer to MRL and the safety of raw materials and food.

Author Response

Dear reviewer,

Our team of authors wish to express our thanks for taking the time to conduct a review of our manuscript entitled "Effect of cephalosporin antibiotics on the activity of yoghurt cultures", as well as for your insightful comments, questions and recommendations.

Yours sincerely

Pavlina Navratilova